# DEEP RL FOR BLOOD GLUCOSE CONTROL: LESSONS, CHALLENGES, AND OPPORTUNITIES

## ABSTRACT

Individuals with type 1 diabetes (T1D) lack the ability to produce the insulin their bodies need. As a result, they must continually make decisions about how much insulin to self-administer in order to adequately control their blood glucose levels. Longitudinal data streams captured from wearables, like continuous glucose monitors, can help these individuals manage their health, but currently the majority of the decision burden remains on the user. To relieve this burden, researchers are working on closed-loop solutions that combine a continuous glucose monitor and an insulin pump with a control algorithm in an 'artificial pancreas.' Such systems aim to estimate and deliver the appropriate amount of insulin. Here, we develop reinforcement learning (RL) techniques for automated blood glucose control. Through a series of experiments, we compare the performance of different deep RL approaches to non-RL approaches. We highlight the flexibility of RL approaches, demonstrating how they can adapt to new individuals with little additional data. On over 21k hours of simulated data across 30 patients, RL approaches outperform baseline control algorithms (increasing time spent in normal glucose range from 71% to 75%) without requiring meal announcements. Moreover, these approaches are adept at leveraging latent behavioral patterns (increasing time in range from 58% to 70%). This work demonstrates the potential of deep RL for controlling complex physiological systems with minimal expert knowledge.

## 1 INTRODUCTION

Type 1 diabetes (T1D) is a chronic disease affecting 20-40 million people worldwide (You & Henneberg, 2016), and its incidence is increasing (Tuomilehto, 2013). People with T1D cannot produce insulin, a hormone that signals cells to uptake glucose in the bloodstream. Without insulin, the body must metabolize energy in other ways that, when relied on repeatedly, can lead to life-threatening conditions (Kerl, 2001). Tight glucose control improves both short- and long-term outcomes for people with diabetes, but can be difficult to achieve in practice (Diabetes Control and Complications Trial Research Group, 1995). Typically, blood glucose is controlled by a combination of basal insulin (to control baseline blood glucose levels) and bolus insulin (to control glucose spikes after meals). To control blood glucose levels, individuals with T1D must continually make decisions about how much basal and bolus insulin to self-administer. This requires careful measurement of glucose levels and carbohydrate intake, resulting in at least 15-17 data points a day. If the individual uses a continuous glucose monitor (CGM), this can increase to over 300 data points, or a blood glucose reading every 5 minutes (Coffen & Dahlquist, 2009).

Combined with an insulin pump, a wearable device that automates the delivery of insulin, CGMs present an opportunity for closed-loop control. Such a system, known as an 'artificial pancreas' (AP), automatically anticipates the amount of required insulin and delivers the appropriate dose. This would be life-changing for individuals with T1D. For many years, researchers have worked towards the creation of an AP for blood glucose control (Kadish, 1964; Bequette, 2005; Bothe et al., 2013). Though the technology behind CGMs and insulin pumps has advanced, there remains significant room for improvement when it comes to the control algorithms (Bothe et al., 2013; Pinsker et al., 2016). Current approaches often fail to maintain sufficiently tight glucose control and require meal announcements.

In this work, we investigate the utility of a deep reinforcement learning (RL) based approach for blood glucose control (Bothe et al., 2013). Deep RL is particularly well-suited for this task because it: i) makes minimal assumptions about the structure of the underlying process, allowing the same system to adapt to different individuals or to changes in individuals over time, ii) can learn to leverage latent patterns such as regular meal times, and iii) scales well in the presence of large amounts of training data. Finally, it can take advantage of existing FDA-approved simulators for model training. Despite these potential benefits, we are not aware of any previously published work that has rigorously explored the feasibility of deep RL for blood glucose control.

While the opportunities for learning an AP algorithm using deep RL are clear, there are numerous challenges in applying standard techniques to this domain. First, there is a significant delay between actions and outcomes; insulin can affect glucose levels hours after administration and this effect can vary significantly across individuals. Without encoding knowledge of patient-specific insulin dynamics, learning the long-term impact of insulin is challenging. Second, compared to tasks that rely on a visual input or are given ground truth state, this task must rely on a noisy observed signal that requires significant temporal context to accurately interpret. Third, because of fluctuations throughout the day and even the week, tight blood glucose control requires small changes in insulin during the day, in addition to large doses of insulin to control glucose spikes. Fourth, unlike game settings where one might have the ability to learn from hundreds of thousands of hours of gameplay, to be practical, any learning approach to blood glucose control must be able to achieve strong performance using only a limited number of days of patient-specific data. Finally, controlling blood glucose levels is a safety-critical application. This sets the bar high from an evaluation perspective. It is unsafe to deploy a system without a human-in-the-loop if there is even a small probability of failure.

Given these challenges, this task represents a significant departure from deep RL baselines. Achieving strong performance in this task requires numerous careful design decisions. In this paper, we make significant progress in this regard, presenting the first deep RL approach that surpasses human-level performance in controlling blood glucose without requiring meal announcements. More specifically, we:

- present an input representation that carefully balances encoding action history and recent changes in our state space,
- propose a patient-specific action space that is amenable to both small and large fluctuations of insulin,
- introduce an augmented reward function, designed to balance the risk of hypo- and hyper-glycemia while drastically penalizing unsafe performance,
- rigorously test the ability of a recurrent architecture to learn from the noisy input, and
- demonstrate how policies can transfer across individuals, dramatically lowering the amount of data required to achieve strong performance while improving safety.

Further, we build on an open-source simulator and make all of our code publicly available [1]. This work can help to build the foundation of a new, tractable, and societally important benchmark for the RL community.

## 2 BACKGROUND AND RELATED WORKS

In recent years, researchers have started to explore RL in healthcare. Examples include matching patients to treatment in the management of sepsis (Weng et al., 2017; Komorowski et al., 2018) and mechanical ventilation (Prasad et al., 2017). In addition, RL has been explored to provide contextual suggestions for behavioral modifications (Klasnja et al., 2019). Despite its success in other problem settings, RL has yet to be fully explored as a solution for a closed-loop AP system (Bothe et al., 2013). RL is a promising solution to this problem, as it is well-suited to learning complex behavior that readily adapts to changing domains (Clavera et al., 2018). Moreover, unlike many other disease settings, there exist credible simulators for the glucoregulatory system (Visentin et al., 2014). The presence of a credible simulator alleviates many common concerns of RL applied to problems in health (Gottesman et al., 2019).

---

[1]Currently hosted at https://tinyurl.com/y6e2m68b, after review a formal code release will be made available on the authors github account

## 2.1 CURRENT AP ALGORITHMS AND RL FOR BLOOD GLUCOSE CONTROL

Among recent commercial AP products, proportional-integral-derivative (PID) control is one of the most common backbones (Trevitt et al., 2015). The simplicity of PID controllers make them easy to use, and in practice they achieve strong results. For example, the Medtronic Hybrid Closed-Loop system, one of the few commercially available, is built on a PID controller (Garg et al., 2017; Ruiz et al., 2012). In this setting, a hybrid closed-loop controller automatically adjusts basal insulin rates, but still requires human-directed insulin boluses to adjust for meals. The main weakness of PID controllers, in the setting of blood glucose control, is their reactivity. As they only respond to current glucose values (including a derivative), often they cannot respond fast enough to meals to satisfactorily control postprandial excursions without meal announcements (Garg et al., 2017). And, without additional safety modifications can overcorrect for these spikes, triggering postprandial hypoglycemia (Ruiz et al., 2012). In contrast, we hypothesize that an RL approach will be able to leverage patterns associated with meal times, resulting in better policies that do not require meal announcements. Moreover, such approaches can take advantage of existing simulators for training and evaluation (described in more detail later).

Previous work has examined the use of RL for different aspects of blood glucose control. Weng et al. (2017) use RL to learn policies that set blood glucose targets for septic patients, but do not learn policies to achieve these targets. Several recent works have investigated the use RL to adapt existing insulin treatment regimes (Ngo et al., 2018; Oroojeni Mohammad Javad et al., 2015; Sun et al., 2018). In contrast to our setting, in which we aim to learn a closed-loop control policy, this work has focused on a human-in-the-loop setting, in which the goal is to learn optimal correction factors and carbohydrate ratios that can be used in the calculation of boluses. Most similar to our own work, De Paula et al. (2015) develop a kernelized Q-learning framework for closed loop glucose control. They make use of Bayesian active learning for on-the-fly personalization. This work tackles a similar problem to our own, but uses a simple two-compartment model for the glucoregulatory system and a fully deterministic meal routine. In our simulation environment, we found that such a Q-learning did not lead to satisfactory closed-loop performance and instead we examine deep actor-critic algorithms for continuous control.

## 2.2 GLUCOSE MODELS AND SIMULATION

Models of the glucoregulatory system have long been important to the development and testing of an AP (Cobelli et al., 1982). Current models are based on a combination of rigorous experimentation and expert knowledge of the underlying physiological phenomena. Typical models consist of a multi-compartment model, with various sources and sinks corresponding to physiological phenomena, involving often dozens of patient-specific parameters. One such simulator, the one we use in our experiments, is the UVA/Padova model (Kovatchev et al., 2009). Briefly, this simulator models the glucoregulatory system as a nonlinear multi-compartment system, where glucose is generated through the liver and absorbed through the gut and controlled by externally administered insulin. A more detailed explanation can be found in (Kovatchev et al., 2009). We use an open-source version of the UVA/Padova simulator that comes with 30 virtual patients, each of which consists of several dozen parameters fully specifying the glucoregulatory system (Xie, 2018). The patients are divided into three classes: children, adolescents, and adults, each with 10 patients. While the simulator we use includes only 10 patients per class, there is a wide range of patient types among each class, with ages ranging from 7-68 years and recommended daily insulin from 16 units to over 60.

## 3 METHODS

The use of deep RL for blood glucose control presents several challenges. Through extensive experimentation, we found that the choice of state representation, action space, and reward function have significant impact on training and validation performance. Additionally, the high sample complexity of standard RL approaches for continuous control tasks can make the application of these methods in real-world settings infeasible. We address these challenges in turn, developing a learning pipeline that achieves strong performance across 30 different patients with the same architecture and hyperparameters without requiring meal announcements. Finally, we demonstrate how such policies can be transferred across patients in a data-efficient manner.

We begin by formalizing the problem. We then describe deep RL approaches that vary in terms of architecture and state representation, and present several baselines: an analogue to human-control in the form of a basal-bolus controller and variants on a PID controller.

## 3.1 PROBLEM SETUP

We frame the problem of blood glucose control as a Markov decision process (MDP) consisting of the 4-tuple $(S, A, P, R)$. Our precise formulation of this problem varies depending on the method and setting. Here, we describe the standard formulation, and explain further differences as they arise. States $\mathbf{s}_t \in S$ consist of the previous 4 hours of blood glucose and insulin data at the resolution of 5-minute intervals: $\mathbf{s}_t = [\mathbf{b}^t, \mathbf{i}^t]$ where:

$$\mathbf{b}^t = [b_{t-47}, b_{t-46}, \dots b_t], \mathbf{i}^t = [i_{t-47}, i_{t-46}, \dots i_t]$$

and $b_t \in \mathcal{N}_{40:400}$, $i_t \in \mathcal{R}_{\geq 0}$, $t \in \mathcal{N}_{1:288}$ and represents a time index for a day at 5-minute resolution. We systematically explored history lengths between 1 and 24 hours as input. After tuning on the validation data, we found that 4 hours struck a good balance between time to convergence and strong performance. We use an update resolution of 5 minutes to mimic the sampling frequency of many common continuous glucose monitors.

Actions $a_t \in \mathcal{R}_{\geq 0}$ are real positive numbers, denoting the size of the insulin bolus in medication units. We experimented with numerous discretized action spaces (as is required by Q-learning approaches), but given the extreme range of insulin values required to achieve good performance, designing an action space where exploration was feasible proved impractical (as a mistimed bolus can be extremely dangerous).

The transition function $P$, our simulator, consists of two elements: i) $G : (a_t, c_t) \to (b_{t+1}, i_{t+1})$, where $c_t \in \mathcal{R}_{\geq 0}$ is the amount of carbohydrates input at time $t$ and $G$ is a model of the glucoregulatory system, its behavior is defined in accordance with the UVA/Padova simulator (Kovatchev et al., 2009), ii) $M : t \to c_t$ is the meal schedule, and is defined in **Appendix A.1**. Note that these equations refer to our simulator, they do not impact our model specifications.

The reward function $R$ is defined as negative risk $-risk(b_t)$ where $risk$ is the asymmetric blood glucose risk function defined as:

$$risk(b) = 10 * (1.509 * \log(b)^{1.084} - 5.381)$$

shown in **Figure 1**, and is an established tool for computing glucose risk (Clarke & Kovatchev, 2009). We add an additional termination penalty to this reward function, where trajectories that enter dangerous blood glucose regimes (blood glucose levels less than 10 or more than 1,000 mg/dL) receive a reward of $-1e5$. We investigated other reward functions, such as time in range or distance from a target blood glucose value, or risk without a termination penalty, but found that optimizing for the proposed reward function consistently led to better control. In particular, it led to control schemes that were less prone to bouts of extreme hypoglycemia, as these trajectories were penalized much more heavily than occasional hyperglycemia. Avoiding extreme hypoglycemia was one of the major challenges we encountered in applying deep RL to blood glucose control.

## 3.2 SOFT ACTOR CRITIC

Our RL controller is trained using the Soft Actor Critic algorithm (Haarnoja et al., 2018). This algorithm is a natural choice for an AP algorithm, as it has been shown to be a reasonably sample efficient and well-performing algorithm when learning continuous control policies. This approach, a member of the Actor-Critic family of algorithms, trains a stochastic policy network (or actor) parameterized by $\phi$ via to maximize the Maximum Entropy RL objective function:

$$J(\pi) = \sum_0^T \mathbb{E}_{(\mathbf{s}_t, a_t) \sim P(s_{t-1}, \pi_\phi(s_{t-1}))} [R(\mathbf{s}_t, a_t) + \alpha H(\pi_\phi(\cdot|\mathbf{s}_t))],$$

where the entropy regularization term, $H$, added to the expected cumulative reward improves exploration and robustness. This objective function is optimized by minimizing the KL divergence between the action distribution and the distribution induced by state-action values:

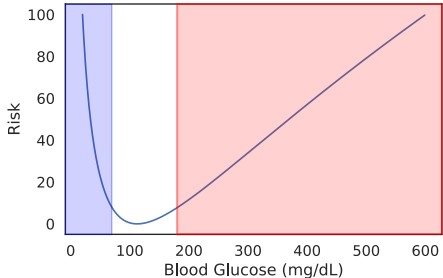

Figure 1: The risk function proposed in (Clarke & Kovatchev, 2009). The mapping between blood glucose values (in mg/dL, x-axis) and Risk values (y-axis). The hypo- and hyperglycemic thresholds are shown as shaded regions. The risk at the threshold of each region is approximately 7.75.

$$J_\pi(\phi) = \mathbb{E}_{\mathbf{s}_t \sim \mathcal{D}} \left[ \mathrm{D}_{\mathrm{KL}} \left( \pi_\phi \left( \cdot | \mathbf{s}_t \right) \| \frac{\exp \left( Q_\theta \left( \mathbf{s}_t, \cdot \right) \right)}{Z_\theta \left( \mathbf{s}_t \right)} \right) \right]$$

where $\mathcal{D}$ is a replay buffer containing previously seen $(\mathbf{s}_t, \mathbf{a}_t, r_t, \mathbf{s}_{t+1})$ tuples, $Z_\theta$ is a partition function, and $Q_\theta$ is the state-action value function parameterized by a neural network (also called a critic) and trained by minimizing the temporal difference loss:

$$J_Q(\theta) = \mathbb{E}_{(\mathbf{s}_t, \mathbf{a}_t) \sim \mathcal{D}} \left[ \frac{1}{2} \left( Q_\theta \left( \mathbf{s}_t, \mathbf{a}_t \right) - \hat{Q} \left( \mathbf{s}_t, \mathbf{a}_t \right) \right)^2 \right],$$

$$\hat{Q} \left( \mathbf{s}_t, \mathbf{a}_t \right) = r \left( \mathbf{s}_t, \mathbf{a}_t \right) + \gamma \mathbb{E}_{\mathbf{s}_{t+1} \sim p} \left[ V_{\bar{\psi}} \left( \mathbf{s}_{t+1} \right) \right].$$

$V_\psi$ is the soft value function parameterized by a third neural network, trained to minimize:

$$J_V(\psi) = \mathbb{E}_{\mathbf{s}_t \sim \mathcal{D}} \left[ \frac{1}{2} \left( V_\psi \left( \mathbf{s}_t \right) - \mathbb{E}_{\mathbf{a}_t \sim \pi_\rho} \left[ Q_\theta \left( \mathbf{s}_t, \mathbf{a}_t \right) - \log \pi_\phi \left( \mathbf{a}_t | \mathbf{s}_t \right) \right] \right)^2 \right],$$

and $V_{\bar{\psi}}$ is the running exponential average of the weights of $V_\psi$ over training (a continuous variant of the hard target network replication in (Mnih et al., 2015)). Additional details of this approach, including the gradient calculations, are given in (Haarnoja et al., 2018). Note that we replace the MSE temporal difference loss with Huber loss, as we find this improves convergence.

### 3.2.1 RECURRENT ARCHITECTURE

Our proposed approach takes as input only the past 4 hours of CGM and insulin data, mimicking real-world applications without human input (*i.e.*, no meal announcements). To extract useful state information from the noisy CGM and insulin history, we parameterize $Q_\theta$, $V_\psi$, and $\pi_\phi$ using GRU networks (Cho et al., 2014), as these types of architectures have successfully been used to model to blood glucose data in the past (Fox et al., 2018; Zhu et al., 2018). The GRU in $\pi_\phi$ maps states to a normal distribution $N(\mu, \log(\sigma))$, from which actions are sampled. Our GRU networks are two layers and have a hidden state size of 128, followed by a fully-connected output layer.

**Patient-Specific Action Space.** After the network output layer, actions are squashed using a *tanh* function. Note that this results in half the action space corresponding to negative values, which we interpret as administering no insulin. This encourages sparse insulin utilization and increases learning speed. We scaled these outputs by a parameter $\omega_b = 43.2 * bas$, where $bas$ is the suggested basal insulin rate (which varies per-person). This scaling ensures that the maximum amount of insulin delivered over a five minute interval is roughly equal to a normal meal bolus, and is derived using the average ratio of basal to bolus insulin in a day (Kuroda et al., 2011). This scaling was an important addition to stabilize training with a continuous action space.

**Efficient Policy Transfer.** Given that one of the main disadvantages of deep RL approaches is their sample efficiency, we sought to explore transfer learning techniques that could allow networks trained from scratch to be efficiently transferred to new patients. We refer to our method trained from scratch as SAC-GRU, and the transfer approach as SAC-GRU-Trans. For SAC-GRU-Trans, we initialize

$Q_\theta, V_\psi, \pi_\phi$ for each class of patients (children, adolescents, and adults) using fully trained networks from one randomly selected member of that source population (*e.g.*, Child/Adolescent/Adult 1). We then fine-tune these networks on data collected from the target patient. This provides a simple approach for training policies with potentially far less data per-patient.

### 3.2.2 ORACLE ARCHITECTURE

A deep RL approach to learning AP algorithms requires that: i) the representation learned by the network contain sufficient information to control the system, and ii) an appropriate control algorithm be learned through interaction with the glucoregulatory system. As we are working with a simulator, we can explore the difficulty of task (ii) in isolation, by replacing the state $s_t$ with the ground-truth state of the simulator $s_t^*$, a 13-dimensional vector with real-valued elements representing glucose, carbohydrate, and insulin values in different compartments of the body. Though unavailable in real-world settings, this representation decouples the problem of learning a policy from that of learning a good state representation. Here, $Q_\theta$, $V_\psi$, and $\pi_\phi$ are fully-connected with two hidden layers, each with 256 units. The network uses ReLU nonlinearities and BatchNorm (Ioffe & Szegedy, 2015).

### 3.3 BASELINES

We examine three baseline methods for control: basal-bolus (BB), PID control, and PID with meal announcements. BB reflects typical human-in-the-loop control strategies, PID reflects a common control strategy used in preliminary fully closed loop AP applications, PID with meal announcements is based on current AP technology, and requires regular human intervention.

### 3.3.1 BASAL-BOLUS BASELINE

This baseline is designed to mimic human control and is typical of how an individual with T1D currently controls their blood glucose. In this setting, we modify the standard state representation $s_t$ to include a carbohydrate signal and a cooldown signal (explained below), and to remove all non-current measurements $s_t = [b_t, i_t, c_t, cooldown]$. Note that the inclusion of a carbohydrate signal, or meal announcement, places the burden of providing accurate and timely estimates of meals on the person with diabetes. Each virtual patient in the simulator comes with the parameters necessary to calculate optimal basal insulin rate $bas$, a correction factor $CF$, and carbohydrate ratio $CR$. These three parameters, together with a glucose target $b_g$ define a clinician-recommended policy $\pi(s_t) = bas + (c_t > 0) * (\frac{c_t}{CR} + cooldown * \frac{b_t - b_g}{CF})$ where $cooldown$ is 1 if there have been no meals in the past three hours, otherwise it is 0. This ensures that each meal is only corrected for once, otherwise meals close in time could lead to over-correction and hypoglycemia. These three parameters can be estimated by endocrinologists using previous glucose and insulin information (Walsh et al., 2011). The parameters for our virtual patient population are given in **Appendix A.2**.

### 3.3.2 PID BASELINE

Variants of PID controllers are already used in commercial AP applications (Garg et al., 2017). A PID controller operates by setting the control variable, here $a_t$, to the weighted combination of three terms $a_t = k_P P(b_t) + k_I I(b_t) + k_D D(b_t)$ such that the process variable $b_t$ (where $t$ is again the time index) remains close to a specified setpoint $b_g$. The terms are calculated as follows: i) the proportional term $P(b_t) = \max(0, b_t - b_g)$ increases the control variable proportionally to the distance from the setpoint, ii) the integral term $I(b_t) = \sum_{j=0}^{t}(b_j - b_g)$ acts to correct long-term deviations from the setpoint, and iii) the derivative term $D(b_t) = |b_t - b_{t-1}|$ acts to control a basic estimate of the future, here approximated by the rate of change. The set point and the weights (also called gains) $k_P, k_D, k_I$ are hyperparameters. To compare to the strongest PID controller possible, we tuned these hyperparameters extensively using multiple iterations of grid-search with exponential refinement per-patient. Our final parameters are presented in **Appendix A.3**

**PID with Meal Announcements.** This baseline, which is designed to be similar to commercially available hybrid closed loop systems (Garg et al., 2017; Ruiz et al., 2012), combines the BB with the PID algorithm into a control algorithm which we call PID with meal announcements (PID-MA). During meals, insulin boluses are calculated and applied as in the BB approach, but instead of using a predetermined fixed basal insulin rate, the PID algorithm controls the basal rate, allowing

for adaptation between meals. We similarly tune the gain parameters for the PID algorithm using sequential grid search with exponential refinement.

### 3.4 Experimental Setup & Evaluation

To measure the utility of deep RL for the task of blood glucose control, we learned policies using the approaches described above, and tested these policies on simulated data with different random seeds across 30 different individuals.

We trained our models (from scratch) for 300 epochs (batch size 256, epoch length 20 days) with an experience replay buffer of size 1e6 and a discount factor of 0.99. While our RL algorithms typically achieved reasonable performance within the first 20-50 epochs of training, we found that additional training was required for stable performance across a range of potential meal schedules. We also found that a sufficient epoch length was critical for learning a stable controller. Too small of an epoch length leads to overly dangerous control policies. We trained our RL models using automatic entropy tuning and sampling actions for exploration (Haarnoja et al., 2018). We optimized the $Q$, $V$ and $\pi$ losses using Adam with a learning rate of $10^{-3}$. All network hyperparameters were optimized on training seeds on a subset of the virtual patients. Our networks were initialized using PyTorch defaults. When fine-tuning models transferred across patients we then train for 50 epochs with a learning rate of $10^{-4}$. We experimented with different loss functions, and found that a Huber loss consistently performed better than a MSE loss. We hypothesize that this is because it more robust to temporary spikes in risk due to meals. All of our code will be made publicly available to allow for replication. For the purpose of anonymous peer-review, we have made our code available on an anonymous google drive account [2].

We measured the performance (mean risk) of the policy networks on 10 days of validation data after each epoch. After training over all epochs, we evaluated using the model parameters from the best epoch as determined by the validation run. Our method of choosing the best validation run led to significant changes in performance. We optimized validation selection using a second pool of validation data generated for one individual. We select our epoch by first filtering out runs that could not control blood glucose within safe levels over the validation run (glucose between 30-1000 mg/dL). We then selected among the remaining epochs the one that achieved the lowest mean risk. On the separated test set 10 days in length (for each patient), we evaluated each control algorithm using i) mean risk, ii) % time spent euglycemic, and iii) % time hypo/hyperglycemic. The random seeds controlling noise and meals in the environment were different between training, validation, and test runs. To account for variability in these seeds, we ran the pipeline three times for each patient.

### 4 Experiments and Results

We investigate the performance of several different classes of policies under different settings. We compare the performance of the BB controller, the PID with and without meal announcements, and the SAC approaches with the Oracle and learned representation across the thirty virtual patients. In follow-up experiments, we demonstrate the efficiency of transferring learned policies across patients relative to training from scratch, and examine the ability of the RL approach to leverage latent behavioral patterns.

**Baseline Models vs. SAC.** Results comparing the BB, PID, and PID-MA baselines to the SAC-GRU-Trans network are given in **Figure 2**. Each point represents a different policy, resulting from a different initialization. Despite the variation across individuals, a clear tread emerges: closed-loop control algorithms that can deliver frequent small doses of insulin can significantly outperform a BB controller. This suggests that, in addition to relieving decision burden, AP systems could lead to overall better blood glucose control. The SAC-GRU-Trans achieved a significantly lower risk than the pure PID (using an independent t-test, $p < 10^{-4}$), and matched the performance of the PID-MA algorithm without requiring meal announcements.

The mean risk for many individuals is above the risk threshold for hyper/hypoglycemia of 7.75. This is far from the optimal level of control. However, it is not the case that all time is spent hypo/hyperglycemic. Across patients, approximately 60-80% of time is spent euglycemic, compared

---

[2]https://tinyurl.com/y6e2m68b

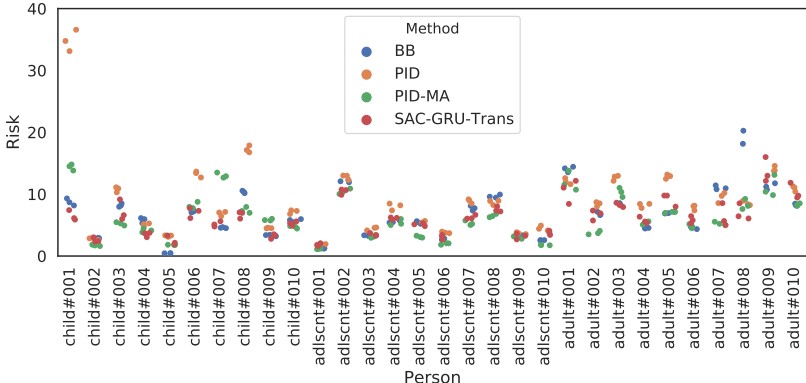

Figure 2: The mean risk over 10 days from different methods applied to different simulated patients. Each point corresponds to a different random seed, that controls initialization, the meal schedule, and randomness in training. On average, the SAC and PID-MA methods perform best.

Table 1: Mean risk, and percent of time Eu/Hypo/Hyperglycemic over 10 days of simulation, 3 runs each for 30 patients ($\pm$ standard deviation). Hybrid and Non-closed loop approaches (requiring meal announcements) are indicated with *. The approach with the best average score is underlined, the best approach that does not require meal announcements is bolded. Among the approaches that do not require meal announcements, SAC-GRU-Trans achieves the lowest risk and most time Euglycemic.

|  | Risk | Euglycemia (%) | Hypoglycemia (%) | Hyperglycemia (%) |
|---|---|---|---|---|
| BB* | $8.99 \pm 21.83$ | $72.61 \pm 86.02$ | $7.76 \pm 11.73$ | $19.63 \pm 11.95$ |
| PID | $9.10 \pm 6.14$ | $71.03 \pm 11.30$ | $\mathbf{2.29 \pm 3.52}$ | $26.67 \pm 11.07$ |
| PID-MA* | $6.16 \pm 3.36$ | $76.07 \pm 13.37$ | $6.70 \pm 7.30$ | $17.23 \pm 10.30$ |
| SAC-Oracle* | $\underline{3.21 \pm 2.03}$ | $\underline{86.52 \pm 9.40}$ | $\underline{1.42 \pm 2.11}$ | $\underline{12.07 \pm 8.32}$ |
| SAC-GRU | $16.77 \pm 54.09$ | $71.43 \pm 17.17$ | $9.66 \pm 10.16$ | $18.91 \pm 14.17$ |
| SAC-GRU-Trans | $\mathbf{6.14 \pm 2.86}$ | $\mathbf{75.04 \pm 11.12}$ | $6.80 \pm 4.55$ | $\mathbf{18.15 \pm 9.14}$ |

with $52\% \pm 19.6\%$ observed in real human control (AyanoTakahara et al., 2015). If insulin is not given well in advance of meals, glucose can increase significantly for a brief period of time, leading to elevated average/mean risk. This skews the distribution of risk towards hyperglycemia and therefore increased risk.

We examine additional models and metrics in the results presented in **Table 1**. We observe that the SAC-Oracle approach is the best across all metrics. This demonstrates an advantage of RL-based control schemes, when given additional information it is simple to improve performance. Among more realistic approaches, PID-MA and SAC-GRU-Trans are comparable in terms of performance. Interestingly, SAC-GRU performs worse on average compared to SAC-GRU-Trans. This is due to occasional catastrophic errors in the policy trained from scratch, where final performance is dangerously poor (5 runs across 3 patients resulted glucose traces with a mean risk of more than 25). The process of transferring and fine-tuning policies eliminates these all of these failures.

**Efficient Policy Transfer.** While SAC-GRU-Trans achieves stronger performance than SAC-GRU with less patient-specific data, it still requires a large amount for any one individual in a non-simulation setting. In **Figure 3a**, we show the average policy performance by epoch of target training. We see that, in the median case, far less training is required to achieve good performance. For half the individuals, we outperform the PID controller within 3 epochs of fine-tuning (or 60 days). However, without a significant number of update epochs, the learned policies may still result in catastrophic failures which lowers mean performance. With our current approach approximately 10 epochs of updating are required to eliminate these catastrophic failures. Notably, we find that policies successfully transfer across dissimilar patients after limited fine tuning. For example, a model trained on the 61 year old adult#001, slightly outperforms the model trained from scratch on the 26 year old adult#006 when applied to adult#006(mean risk 5.83 vs. 5.99). Similarly, applied to adult#005, the fine-tuned adult#001 model outperformance the from-scratch adult#005 model (mean risk 9.17

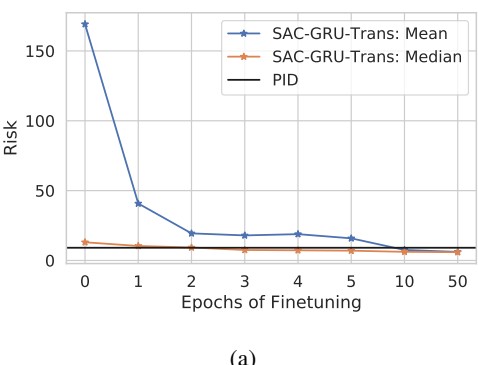 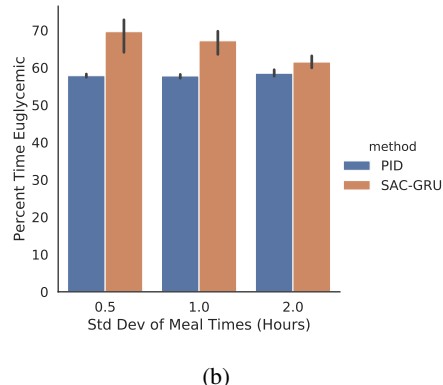

(a)                                                          (b)

Figure 3: a) The impact of fine-tuning SAC-GRU-Trans; performance reported across all patients. While median performance rapidly surpasses the PID (within 3 epochs of fine-tuning), it takes 10 epochs for mean performance to surpass the PID due to catastrophic failures after initial transfer. b) Percent of time Euglycemic over 10 days for Adult 1 using different meal schedules. As meal times become more predictable (lower standard deviation), SAC-GRU performs better.

vs. 13.00). These patients differ not only in age (61 v. 55) but also in terms of average amounts of required daily insulin (50 units versus 68 units). We are encouraged by the ease with which our learned policies transfer to different individuals suggests the general applicability of our approach.

**Ability to Adapt to Meals** We hypothesize that one of the potential advantages of RL is its ability to exploit underlying behavioral patterns. To investigate this potential benefit, we explored changing the meal schedule generation procedure outlined in **Algorithm 1** for Adult 1. We removed the 'snack' meals (those with occurrence probabilities of 0.3) and set all meal occurrence probabilities to 1 and meal amount standard deviations to 0 (*i.e.*, each day Adult 1 consumes an identical set of meals). We then evaluated both the PID model and the SAC-GRU model on 3 variations of this environment, characterized by the standard deviation of the meal times (either 0.5, 1, or 2 hours). This tests the ability of each method to take advantage of latent patterns in the environment. The results are presented in **Figure 3b**. We observe that, while SAC-GRU achieves lower risk than PID under all settings, the difference becomes more pronounced as the standard deviation of meal times becomes smaller (and thus meal timing becomes more predictable). This demonstrates that SAC-GRU is better able to leverage latent meal patterns.

**Learning with and Corrupting Meal Announcements** One of the major advantages of our proposed approach is its ability to achieve strong performance without meal announcements. Not only does requiring meal announcements impose a burden on individuals with diabetes, but accurately estimating carbohydrates from a meal can be challenging. Multiple studies have found systematic errors in carb counting among adults (Brazeau et al.), adolescents and children (Deeb et al., 2017), which can significantly impact glycemic control. To investigate the effect of realistic carb-miscounting on our baselines, we added normally distributed noise (with mean 0 and standard deviation equal to 20% of the meal size, in keeping with past literature (Reiterer et al.)), to the carbohydrate size estimates in the BB controller. We found this noise greatly increased mean risk, from 8.99 to 13.51. This suggests that, in practice, our proposed approach could improve glycemic control over a BB controller beyond what is suggested in **Table 1**, especially in the presence of erroneous carbohydrate estimates.

For completeness, we examined how our proposed approach might benefit from meal announcements. We included the number carbohydrates consumed at each time step $t$ as an additional input channel to our actor and critic networks trained from scratch on adult#001 on three random seeds. For each training seed, we evaluated on 100 different test seeds, and compared against the performance of our baseline SAC-GRU on that seed. We observe including a meal announcement decreases mean risk from 11.49 to 9.85 (as an additional comparison, on adult#001 PID-MA achieves a mean risk of 10.53). This suggests that, while our approach does not require meal announcements to achieve strong performance, it could still benefit from them.

## 5 DISCUSSION AND CONCLUSION

In this work, we develop and explore deep RL algorithms to learn automated blood glucose control policies. When given information about the ground truth state, a soft actor-critic (SAC-Oracle) convincingly outperformed baseline approaches. Without access to the ground-truth state, or even meal announcements, a recurrent SAC outperformed both the BB and PID baselines, matching the performance of a PID with meal announcements. Moreover, this approach was able to significantly improve performance in the presence of a predictable meal schedule.

The use of policy transfer was found to be important in stabilizing performance for the SAC-GRU. Beyond the performance of the learned policies, across our experiments, we found that thousands of days of simulation data were required when training our deep approaches from scratch. However, by transferring policies across individuals and fine-tuning, we were able to learn with far less data (and indeed, such transfer performs better on average than training from scratch).

While these results are encouraging, there are several limitations. First, our results are based on simulation. While the simulator in question is a highly credible one, it may not adequately capture variation across patients or changes in the glucoregulatory system over time. However, as an FDA-approved substitute for animal trials (Kovatchev et al., 2009), success in this simulator is a nontrivial accomplishment. Second, we define a reward function based on risk. Though optimizing this risk function should lead to tight glucose control, it could lead to excess insulin utilization (as its use is unpenalized). Future work could consider resource-aware variants of this reward. Finally, we emphasize that blood glucose control is a safety-critical application. An incorrect dose of insulin could lead to life-threatening situations. Importantly, the proposed approach, though promising, is not ready for deployment. As shown by the worst-case performance of the SAC-GRU method in **Table 1**, deep approaches can fail catastrophically. We have investigated several ways to minimize such failures, including modifying the reward function and model selection procedure, and these approaches combined with policy transfer successfully avoided such catastrophic failures. However, these empirical results do not guarantee acceptable performance under any possible circumstance. Going forward, there are several approaches that could be investigated to guarantee acceptable worst-case performance. Using the notion of 'shielding' from (Alshiekh et al., 2018), hard limits on insulin informed by blood glucose levels could prevent catastrophic hypoglycemia. Though this, in turn, could limit controller effectiveness in response to rapidly increasing glucose levels. Additionally, approaches that incrementally modify existing safe policies can limit worst-case performance and lead to safer control (Berkenkamp et al., 2017). Despite these limitations, our results clearly demonstrate that deep RL is a promising approach for learning truly closed-loop algorithms for blood glucose control.

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

# A APPENDIX

## A.1 MEAL GENERATION ALGORITHM

---

**Algorithm 1** Generate Meal Schedule

---

**Input:** body weight $w$, number of days $n$
$MealOcc = [0.95, 0.3, 0.95, 0.3, 0.95, 0.3]$
$TimeLowerBounds = [5, 9, 10, 14, 16, 20] * 12$
$TimeUpperBounds = [9, 10, 14, 16, 20, 23] * 12$
$TimeMean = [7, 9.5, 12, 15, 18, 21.5] * 12$
$TimeStd = [1, .5, 1, .5, 1, .5] * 12$
$AmountMean = [0.7, 0.15, 1.1, 0.15, 1.25, 0.15] * w$
$AmountStd = AmountMean * 0.15$
$Days = []$
**for** $i \in [1, \ldots, n]$ **do**
   $M = [0]_{j=1}^{288}$
  **for** $j \in [1, \ldots, 6]$ **do**
    $m \sim Binomial(MealOcc[j])$
    $lb = TimeLowerBounds[j]$
    $ub = TimeUpperBounds[j]$
    $\mu_t = TimeMean[j]$
    $\sigma_t = TimeStd[j]$
    $\mu_a = AmountMean[j]$
    $\sigma_a = AmountStd[j]$
    **if** $m$ **then**
      $t \sim Round(TruncNormal(\mu_t, \sigma_t, lb, ub))$
      $c \sim Round(max(0, Normal(\mu_a, \sigma_a)))$
      $M[t] = c$
    **end if**
  **end for**
   $Days.append(M)$
**end for**

---

| Person | CR | CF | Age | TDI |
|---|---|---|---|---|
| child#001 | 28.62 | 103.02 | 9 | 17.47 |
| child#002 | 27.51 | 99.02 | 9 | 18.18 |
| child#003 | 31.21 | 112.35 | 8 | 16.02 |
| child#004 | 25.23 | 90.84 | 12 | 19.82 |
| child#005 | 12.21 | 43.97 | 10 | 40.93 |
| child#006 | 24.72 | 89.00 | 8 | 20.22 |
| child#007 | 13.81 | 49.71 | 9 | 36.21 |
| child#008 | 23.26 | 83.74 | 10 | 21.49 |
| child#009 | 28.75 | 103.48 | 7 | 17.39 |
| child#010 | 24.21 | 87.16 | 12 | 20.65 |
| adolescent#001 | 13.61 | 49.00 | 18 | 36.73 |
| adolescent#002 | 8.06 | 29.02 | 19 | 62.03 |
| adolescent#003 | 20.62 | 74.25 | 15 | 24.24 |
| adolescent#004 | 14.18 | 51.06 | 17 | 35.25 |
| adolescent#005 | 14.70 | 52.93 | 16 | 34.00 |
| adolescent#006 | 10.08 | 36.30 | 14 | 49.58 |
| adolescent#007 | 11.46 | 41.25 | 16 | 43.64 |
| adolescent#008 | 7.89 | 28.40 | 14 | 63.39 |
| adolescent#009 | 20.77 | 74.76 | 19 | 24.08 |
| adolescent#010 | 15.07 | 54.26 | 17 | 33.17 |
| adult#001 | 9.92 | 35.70 | 61 | 50.42 |
| adult#002 | 8.64 | 31.10 | 65 | 57.87 |
| adult#003 | 8.86 | 31.90 | 27 | 56.43 |
| adult#004 | 14.79 | 53.24 | 66 | 33.81 |
| adult#005 | 7.32 | 26.35 | 52 | 68.32 |
| adult#006 | 8.14 | 29.32 | 26 | 61.39 |
| adult#007 | 11.90 | 42.85 | 35 | 42.01 |
| adult#008 | 11.69 | 42.08 | 48 | 42.78 |
| adult#009 | 7.44 | 26.78 | 68 | 67.21 |
| adult#010 | 7.76 | 27.93 | 68 | 64.45 |

Table 2: Basal-Bolus Parameters

## A.2 BB Parameters

## A.3 PID AND PID-MA PARAMETERS

| | $k_p$ | $k_i$ | $k_d$ |
|---|---|---|---|
| child#001 | -1.00E-05 | -3.68E-08 | -7.59E-04 |
| child#002 | -3.49E-05 | -3.49E-07 | -3.98E-03 |
| child#003 | -6.31E-05 | -2.23E-08 | -1.00E-03 |
| child#004 | -3.49E-05 | -3.49E-07 | -1.00E-03 |
| child#005 | -1.00E-04 | -6.31E-07 | -2.87E-03 |
| child#006 | -6.31E-05 | -2.87E-08 | -1.00E-03 |
| child#007 | -1.00E-05 | -3.49E-07 | -2.51E-03 |
| child#008 | -1.93E-08 | -4.72E-08 | -1.00E-03 |
| child#009 | -1.00E-05 | -3.98E-07 | -1.00E-03 |
| child#010 | -4.98E-07 | -3.49E-07 | -2.09E-03 |
| adolescent#001 | -2.87E-06 | -1.00E-06 | -1.00E-02 |
| adolescent#002 | -5.53E-09 | -4.54E-12 | -1.00E-02 |
| adolescent#003 | -1.00E-04 | -3.49E-07 | -3.98E-03 |
| adolescent#004 | -6.74E-08 | -6.74E-10 | -1.00E-02 |
| adolescent#005 | -4.54E-10 | -2.87E-08 | -1.00E-02 |
| adolescent#006 | -1.93E-08 | -3.49E-06 | -6.31E-03 |
| adolescent#007 | -1.07E-07 | -1.00E-07 | -6.31E-03 |
| adolescent#008 | -6.74E-08 | -8.21E-09 | -1.00E-02 |
| adolescent#009 | -2.35E-07 | -1.00E-06 | -3.98E-03 |
| adolescent#010 | -1.58E-09 | -1.00E-07 | -1.00E-02 |
| adult#001 | -8.32E-05 | -1.00E-07 | -1.00E-02 |
| adult#002 | -3.02E-04 | -1.00E-07 | -1.00E-02 |
| adult#003 | -2.87E-06 | -6.07E-08 | -1.00E-02 |
| adult#004 | -2.87E-05 | -3.49E-07 | -3.98E-03 |
| adult#005 | -1.00E-04 | -1.00E-07 | -1.00E-02 |
| adult#006 | -1.00E-04 | -5.75E-07 | -1.00E-02 |
| adult#007 | -1.35E-06 | -1.58E-07 | -1.00E-02 |
| adult#008 | -4.72E-06 | -1.00E-07 | -1.00E-02 |
| adult#009 | -1.00E-04 | -1.00E-07 | -1.00E-02 |
| adult#010 | -6.31E-05 | -1.00E-07 | -1.00E-02 |

Table 3: PID parameters

| | $k_p$ | $k_i$ | $k_d$ |
|---|---|---|---|
| child#001 | -4.54E-10 | -1.00E-07 | -3.49E-04 |
| child#002 | -1.00E-04 | -5.75E-07 | -2.09E-03 |
| child#003 | -3.49E-05 | -1.00E-07 | -1.00E-03 |
| child#004 | -7.59E-05 | -5.75E-07 | -1.58E-03 |
| child#005 | -1.74E-04 | -4.54E-08 | -3.31E-03 |
| child#006 | -4.54E-10 | -3.49E-07 | -1.00E-03 |
| child#007 | -6.31E-05 | -3.49E-07 | -1.00E-03 |
| child#008 | -2.35E-07 | -1.00E-07 | -1.00E-03 |
| child#009 | -5.53E-09 | -3.31E-07 | -5.75E-04 |
| child#010 | -4.54E-10 | -3.98E-07 | -1.00E-03 |
| adolescent#001 | -4.54E-10 | -1.00E-06 | -1.00E-02 |
| adolescent#002 | -1.00E-04 | -1.00E-07 | -3.49E-03 |
| adolescent#003 | -4.54E-06 | -6.31E-07 | -1.58E-03 |
| adolescent#004 | -6.31E-05 | -3.49E-07 | -3.98E-03 |
| adolescent#005 | -1.00E-05 | -1.00E-06 | -4.37E-03 |
| adolescent#006 | -3.98E-05 | -1.00E-05 | -3.02E-03 |
| adolescent#007 | -3.02E-07 | -3.98E-07 | -3.02E-03 |
| adolescent#008 | -1.00E-04 | -1.00E-07 | -4.37E-03 |
| adolescent#009 | -2.87E-06 | -1.00E-06 | -1.58E-03 |
| adolescent#010 | -8.21E-07 | -1.00E-06 | -1.58E-03 |
| adult#001 | -4.54E-10 | -1.00E-07 | -3.49E-03 |
| adult#002 | -4.54E-10 | -4.54E-12 | -1.00E-02 |
| adult#003 | -4.54E-10 | -5.25E-07 | -1.00E-03 |
| adult#004 | -4.54E-10 | -6.31E-07 | -1.00E-03 |
| adult#005 | -4.54E-10 | -3.88E-09 | -1.00E-02 |
| adult#006 | -4.54E-10 | -1.00E-06 | -2.87E-03 |
| adult#007 | -3.98E-04 | -8.21E-09 | -1.00E-02 |
| adult#008 | -1.00E-04 | -1.00E-07 | -3.49E-03 |
| adult#009 | -4.54E-10 | -6.74E-10 | -1.00E-02 |
| adult#010 | -1.00E-04 | -1.00E-07 | -4.37E-03 |

Table 4: PID-MA parameters

