# OpenReview forum: "Deep RL for Blood Glucose Control: Lessons, Challenges, and Opportunities"
_ICLR.cc/2020/Conference — Reject_

### Official Review · AnonReviewer3 · 2019-10-18
**Official Blind Review #3**

**Rating:** 3

**Review:**

Paper Summary

This paper examines reinforcement learning in the context of blood glucose control to help individuals with type 1 diabetes. The authors show that their methods lead to strong algorithms that can improve artificial pancreas systems. Their results are promising, and, very importantly, do not require meal announcements. The importance of their application is self evident.

Decision

Should their claim to novelty hold up, then the authors have provided evidence that RL can be useful for this important application of glucose control. Overall, the paper is very well written with clear arguments, and the impact of their study for type 1 diabetes is high. However, there are novelty concerns with the proposed methods. The paper needs some work before acceptance.

Additional Feedback

One caveat is that a small search of previous RL methods in blood glucose control did yield some similarly titled papers (please discuss "Reinforcement Learning Algorithm for Blood Glucose Control in Diabetic Patients" by Javad et al), and they were not addressed or compared in this paper. To push this review over the edge, the authors should address these papers in the related work, and discuss how this paper's method compares.

Additionally, the novelty of the actual RL methods is not entirely clear. The authors should very clearly point out their contributions within the methods sections, differentiating between past methods and the proposed one. Most importantly, the authors should write a paragraph-length section at the end of the introduction detailing their proposed methods, with a bullet-point layout of every novel detail. This will help future readers get the gist of the paper more accurately.

Lastly, though the authors addressed the limitations of their dataset in terms of it being a simulation, they should also discuss the sample size being only 10 patients in different age groups. It would be helpful for readers to know how the method will generalize to new patients.


**Experience Assessment:**

I have read many papers in this area.

**Review Assessment: Checking Correctness Of Derivations And Theory:**

I assessed the sensibility of the derivations and theory.

**Review Assessment: Checking Correctness Of Experiments:**

I carefully checked the experiments.

**Review Assessment: Thoroughness In Paper Reading:**

I read the paper at least twice and used my best judgement in assessing the paper.

---

> ### Author Response · Authors · 2019-11-14
> **Response to Reviewer 3**
>
> Thank you for the feedback. We are glad you found our paper well written, our results promising, and that you recognized the potential for impact high.
>
> First, we would like to address a branch of related work, exemplified by “Reinforcement Learning Algorithm for Blood Glucose Control in Diabetic Patients.” In this branch of work, researchers consider a setting that significantly differs from ours. In contrast to our setting in which we aim to learn a closed-loop control policy, past work has focused on a ‘human-in-the-loop’ setting, in which the goal is to learn optimal correction factors and carbohydrate ratios that can be used in the calculation of boluses. Though different, for completeness, we have included a discussion of this branch of work in Section 2.1 of our updated manuscript.
>
> Second, with regard to the specific contributions of our paper, please see our post on the relevance of our problem and our contributions to it. As suggested, we have updated our introduction to include a paragraph with a bulleted list to make our contributions clear.
>
> Finally, we would like to address concerns over the limited number of patients in our simulator. We have performed experiments 30 patients, that represent a broad range of patient characteristics (with ages from 7-68 years and requiring from 15-70 total daily units of insulin). Each methods test set is composed of 900 days blood glucose data.  Moreover, in our transfer experiments, we found that a model trained on one individual could be successfully transferred with little additional data for fine-tuning. In particular, using the model trained for Child/Adolescent/Adult 1, we were able to match or surpass the performance of the patient-specific models for the other patients in the same category. For example, the model trained on the 61 year old Adult 1, after fine-tuning, slightly outperforms the model trained from scratch on the 26 year old Adult 6 (average risk 5.83 vs. 5.99), and another fine-tuned Adult 1 model also outperforms the from-scratch model for Adult 5, who requires 18 units of daily insulin on average, the most of any adult patient (mean risk 9.17 vs. 13.00). For these reasons we believe that our approach will generalize  to additional individuals. We made the limitations regarding the number of simulated individuals and our approaches ability to overcome patient differences more clear in Sections 2.2 and 4 respectively.

---

### Official Review · AnonReviewer1 · 2019-10-22
**Official Blind Review #1**

**Rating:** 3

**Review:**

The paper describes an RL based approach to administer insulin for blood glucose control among  type-1 diabetic patients.  The paper formulates this blood glucose control problem as a closed-loop reinforcement learning problem and demonstrates its effectiveness on data generated from an FDA-approved simulator of glucoregulatory system. Compared to existing approaches, the proposed method can operate without meal announcement by potentially making use of latent meal intake patterns. The authors also demonstrate how a learned policy for one particular subject can be used as initialization to train/fine-tuned the policy of another subject so as to combat the issue of high sample complexity.

Overall, the reviewer finds that the authors provide a reasonable approach to model the blood glucose management problem for type-1 diabetes. Each component of the paper is well-explained and the paper is easy to follow. The algorithmic design adopted in this paper, such as the representation of the problem, the choice of the reward function, and the choice of RL controller, are reasonably justified. While the experiments conducted in this paper are not based on real-world data, the reviewer finds the use of data from an FDA-approved simulator of glucoregulatory system sufficiently convincing for this type of problem. The limitation of the proposed method is also well discussed.

The reviewer has the following concerns:

1. The reviewer views the major contribution of this paper as formulating and solving the glucose management problem as an RL problem.  From a machine learning perspective, the reviewer finds the contribution made in this paper to advance ML methodology very limited as the components used in the algorithmic design of the proposed method are already available in the existing literature. Therefore, the reviewer finds that the paper could be of limited interest to the audience in ICLR while it might be more suitable to the audience of diabetes management.

2. Compared to competing methods reported in this paper, a major characteristic of the proposed method is that it can operate without meal announcements. Methods with meal announcements seem to operate reasonably well compared to the proposed method. Therefore, the authors can consider further justifying why meal announcement is an important bottleneck to alleviate in blood glucose management, which is currently not well explained in the paper. Related to this question, the authors may also consider justifying why making meal announcements more convenient/automated is not a good alternative to handle the blood glucose management problem. It will also be interesting to see how the proposed method will behave when augmented with meal announcements since in reality, the proposed solution might not always be reliable and intervention options like meal announcements could potentially improve the robustness of the solution.


Miscellaneous:
in Section 3.1 of the glucoregulatory system model G, the carbohydrate input $c_t$ seems to be considered as an aspect of the action. Based on the understanding of the reviewer, such an action is a proxy to the meal announcement and is not considered as an input from the user for the deployed policy. For better clarity, the authors can consider reporting the formula of the deployed policy and explain how the quantity $c_t$ is related to this policy.

**Experience Assessment:**

I have published in this field for several years.

**Review Assessment: Checking Correctness Of Derivations And Theory:**

N/A

**Review Assessment: Checking Correctness Of Experiments:**

I assessed the sensibility of the experiments.

**Review Assessment: Thoroughness In Paper Reading:**

I read the paper at least twice and used my best judgement in assessing the paper.

---

> ### Author Response · Authors · 2019-11-14
> **Response to Reviewer 1**
>
> Thank you for taking the time to provide a useful and detailed review. We are glad you found the various components in our pipeline well justified. In this work we aim to advance knowledge in the application of RL techniques, while solving a problem that afflicts a significant portion of society.
>
> First, we would like to clarify the carbohydrate input in Section 3.1. The equations in question are a model used for our training environment (ie: the simulator). Carbohydrate inputs are not used for training or evaluation of our RL approaches. Our learned policies are simply functions mapping from 4 hours of CGM and insulin data to an insulin dose in the next five minutes. We have added a sentence in this section to clarify this.
>
> Second, with respect to concern 1, please see our statement on relevance and contributions. We believe that our work, by laying the foundation for a societally important RL benchmark task, is of interest to the ICLR community.
>
> Finally, with respect to concern 2, meal announcements are an important bottleneck because they require regular and accurate human intervention. While providing accurate meal announcements is not an issue for many, some groups (particularly children and adolescents) can forget to log meals or record meals incorrectly. In Section 4 we have included a new subsection ‘Corrupting and Learning from Meal Announcements’ where we detail two new experiments in line with your requests. In the first experiment, we find that including a realistic amount of carbohydrate error significantly worsens the performance of the BB controller (from average risk 8.99 to 13.51), showing a limitation of methods that rely on meal announcements. In the second experiment, we show that our RL approach can utilize meal announcements to improve performance, improving average risk from 11.49 to 9.85 for adult#001 (below the PID-MA performance of 10.53).

---

### Author Response · Authors · 2019-11-14
**Relevance of the Application and Specific Contributions**

Diabetes affects nearly 1/10 people in the US and is a growing global problem. People with Type 1 diabetes must constantly make decisions about their treatment regimen. Many of these people are children, who are not well-equipped to make such decisions on their own. To this end, for several decades, researchers have sought a closed loop system that does not require meal announcements. A robust solution to this problem would have significant societal implications. Recent advances in deep RL show promise, but from a learning perspective, this problem is particularly challenging, because:
- There is a significant delay between actions and outcomes. Insulin can affect glucose levels hours after administration and this effect can vary significantly across individuals. Without encoding knowledge of patient-specific insulin dynamics, learning the long-term impact of insulin is challenging.
- Compared to tasks that rely on a visual input or are given ground truth state, this task must rely on a noisy signal that requires significant temporal context to interpret.
- With the goal of circumventing the need for meal announcements, we must implicitly infer meals (timing and size). However, the input signal is noisy, and this non-stationary noise can make it difficult to detect if/when a meal has begun.
- Because of fluctuations throughout the day and even the week, tight blood glucose control requires both fine-grained changes in insulin throughout the day and large doses of insulin to control glucose spikes.
- Controlling blood glucose levels is a safety-critical application. This sets the bar high from an evaluation perspective. It is unsafe to deploy a system without a human-in-the-loop with even a small probability of failure.
- Unlike game settings where one might have the ability to learn from hundreds of thousands of hours of gameplay, to be practical, any learning approach to blood glucose control must be able to achieve strong performance using only a limited number of days of patient-specific data.

Due to these challenges, this task represents a significant departure from existing deep RL baselines. Achieving strong performance in this task required numerous careful design decisions, from the reward function to the range of possible actions. In this work, we propose the first deep RL approach for blood glucose control, in which we have:
- Explored different input representations, determining a length of input history that balances the importance of action history with that of recent glucose changes.
- Designed a network architecture and training scheme that is able to reliably learn from noisy glucose trajectories.
Proposed a patient-specific action space that allows for both major and minor changes in insulin level, while being amenable to exploration to allow for learning .
- Augmented a reward function designed to balance the risks of hypo- and hyperglycemia with termination penalties to further ensure learning safe policies.
- Presented an effective and efficient way to transfer policies learned on one individual to another.

Moreover, by adding a preliminary code release to our submission, we have committed to making our solution and problem setting publicly available. Our work can help to build the foundation of a new, tractable, and societally important benchmark for the RL community.

---

### Author Response · Authors · 2019-11-14
**Updates to the Paper**

We thank the reviewers for their useful feedback. Below, we respond to each reviewer, in turn. In addition, we have updated the paper to reflect their suggestions. We also include a separate post, ‘Relevance of the Application and Specific Contributions’ written to respond to issues raised by both reviewers.

 In addition to these changes, we have also worked to improve the clarity of the paper and have worked to strengthen the baselines.  Specifically,  we modified the carbohydrate ratio and correction factor used in our basal bolus controller to bring the approach more in line with standard practice. We found these new parameters significantly improved performance for the basal bolus controller, bringing average risk down from 21.37 to 8.99. These updated parameters did not meaningfully impact the results of the PID-MA (average risk rose from 6.15 to 6.16 after re-tuning the gains). These updated results do not change our conclusions, as we still observe a marked improvement with closed-loop controllers, but they serve to provide a more accurate estimate of human-level performance in blood glucose control. As a result of this change, we have included the new basal-bolus parameters as a new appendix section and have updated the PID-MA parameter table in the appendix.

---

### Decision · Program_Chairs · 2019-12-19

**Decision:**

Reject

**Comment:**

The reviewers all believe that this paper is not yet ready for publication. All agree that this is an important application, and an interesting approach. The methodological novelty, as well as other parts of exposition, involving related work, or further discussion of what this solution means for patients, is right now not completely convincing to reviewers. My recommendation is to work on making sure the exposition best explains the methodology, and making sure this venue is the best for the submitted line of work.